# Application of Artificial Intelligence in Combating High Antimicrobial Resistance Rates

**DOI:** 10.3390/antibiotics11060784

**Published:** 2022-06-08

**Authors:** Ali A. Rabaan, Saad Alhumaid, Abbas Al Mutair, Mohammed Garout, Yem Abulhamayel, Muhammad A. Halwani, Jeehan H. Alestad, Ali Al Bshabshe, Tarek Sulaiman, Meshal K. AlFonaisan, Tariq Almusawi, Hawra Albayat, Mohammed Alsaeed, Mubarak Alfaresi, Sultan Alotaibi, Yousef N. Alhashem, Mohamad-Hani Temsah, Urooj Ali, Naveed Ahmed

**Affiliations:** 1Molecular Diagnostic Laboratory, Johns Hopkins Aramco Healthcare, Dhahran 31311, Saudi Arabia; 2College of Medicine, Alfaisal University, Riyadh 11533, Saudi Arabia; 3Department of Public Health and Nutrition, The University of Haripur, Haripur 22610, Pakistan; 4Administration of Pharmaceutical Care, Al-Ahsa Health Cluster, Ministry of Health, Al-Ahsa 31982, Saudi Arabia; saalhumaid@moh.gov.sa; 5Research Center, Almoosa Specialist Hospital, Alhassa, Al-Ahsa 36342, Saudi Arabia; abbas.almutair@almoosahospital.com.sa; 6Almoosa College of Health Sciences, Alhassa, Al-Ahsa 36342, Saudi Arabia; 7School of Nursing, Wollongong University, Wollongong, NSW 2522, Australia; 8Nursing Department, Prince Sultan Military College of Health Sciences, Dhahran 34313, Saudi Arabia; 9Department of Community Medicine and Health Care for Pilgrims, Faculty of Medicine, Umm Al-Qura University, Makkah 21955, Saudi Arabia; magarout@uqu.edu.sa; 10Specialty Internal Medicine Department, Johns Hopkins Aramco Healthcare, Dhahran 34465, Saudi Arabia; yem.abulhamayel@jhah.com; 11Department of Medical Microbiology, Faculty of Medicine, Al Baha University, Al Baha 4781, Saudi Arabia; mhalwani@bu.edu.sa; 12Immunology and Infectious Microbiology Department, University of Glasgow, Glasgow G1 1XQ, UK; jeehanalostad@gmail.com; 13Microbiology Department, Collage of Medicine, Jabriya 46300, Kuwait; 14Adult Critical Care Department of Medicine, Division of Adult Critical Care, College of Medicine, King Khalid University, Abha 62561, Saudi Arabia; albshabshe@yahoo.com; 15Infectious Diseases Section, Medical Specialties Department, King Fahad Medical City, Riyadh 12231, Saudi Arabia; dr.tarek.sulaiman@gmail.com; 16Basic Medical Sciences, Majmaah University, Majmaah 11952, Saudi Arabia; m.alfonaisan@mu.edu.sa; 17Infectious Disease and Critical Care Medicine Department, Dr. Sulaiman Alhabib Medical Group, Alkhobar 34423, Saudi Arabia; dr_tmusawi@hotmail.com; 18Department of Medicine, Royal College of Surgeons in Ireland-Medical University of Bahrain, Manama 15503, Bahrain; 19Infectious Disease Department, King Saud Medical City, Riyadh 7790, Saudi Arabia; hhalbayat@gmail.com; 20Infectious Disease Division, Department of Medicine, Prince Sultan Military Medical City, Riyadh 11159, Saudi Arabia; mohalsaeed@live.com; 21Department of Pathology and Laboratory Medicine, Sheikh Khalifa General Hospital, Umm Al Quwain 499, United Arab Emirates; mubarak.alfaresi@skgh.ae; 22Department of Pathology, College of Medicine, Mohammed Bin Rashid University of Medicine and Health Sciences, Dubai 505055, United Arab Emirates; 23Molecular Microbiology Department, King Fahad Medical City, Riyadh 11525, Saudi Arabia; salotaibi1@gmail.com; 24Department of Clinical Laboratory Sciences, Mohammed AlMana College of Health Sciences, Dammam 34222, Saudi Arabia; yousefa@machs.edu.sa; 25Pediatric Department, College of Medicine, King Saud University, Riyadh 11451, Saudi Arabia; mtemsah@ksu.edu.sa; 26Department of Biotechnology, Faculty of Life Sciences, University of Central Punjab, Lahore 54000, Pakistan; msurooj9@gmail.com; 27Department of Medical Microbiology and Parasitology, School of Medical Sciences, Universiti Sains Malaysia, Kubang Kerian, Kota Bharu 16150, Kelantan, Malaysia

**Keywords:** antibiotic stewardship, better diagnosis, AMR, global platform, advances, diagnostic microbiology

## Abstract

Artificial intelligence (AI) is a branch of science and engineering that focuses on the computational understanding of intelligent behavior. Many human professions, including clinical diagnosis and prognosis, are greatly useful from AI. Antimicrobial resistance (AMR) is among the most critical challenges facing Pakistan and the rest of the world. The rising incidence of AMR has become a significant issue, and authorities must take measures to combat the overuse and incorrect use of antibiotics in order to combat rising resistance rates. The widespread use of antibiotics in clinical practice has not only resulted in drug resistance but has also increased the threat of super-resistant bacteria emergence. As AMR rises, clinicians find it more difficult to treat many bacterial infections in a timely manner, and therapy becomes prohibitively costly for patients. To combat the rise in AMR rates, it is critical to implement an institutional antibiotic stewardship program that monitors correct antibiotic use, controls antibiotics, and generates antibiograms. Furthermore, these types of tools may aid in the treatment of patients in the event of a medical emergency in which a physician is unable to wait for bacterial culture results. AI’s applications in healthcare might be unlimited, reducing the time it takes to discover new antimicrobial drugs, improving diagnostic and treatment accuracy, and lowering expenses at the same time. The majority of suggested AI solutions for AMR are meant to supplement rather than replace a doctor’s prescription or opinion, but rather to serve as a valuable tool for making their work easier. When it comes to infectious diseases, AI has the potential to be a game-changer in the battle against antibiotic resistance. Finally, when selecting antibiotic therapy for infections, data from local antibiotic stewardship programs are critical to ensuring that these bacteria are treated quickly and effectively. Furthermore, organizations such as the World Health Organization (WHO) have underlined the necessity of selecting the appropriate antibiotic and treating for the shortest time feasible to minimize the spread of resistant and invasive resistant bacterial strains.

## 1. Introduction

The widespread use of antibiotics in clinical practice has not only resulted in drug resistance but has also increased the threat of super-resistant bacteria emergence. Pakistan is one of the countries that have a high rate of AMR and little healthcare expertise and assistance to tackle it, which raises questions about high AMR rates. Alexander Fleming’s discovery of penicillin in 1928 marked the beginning of the modern age of antibiotics [1]. Since then, antibiotics have saved the lives of many individuals suffering from bacterial and fungal infections. However, the widespread use of antibiotics in clinical practice has resulted in drug resistance, in addition to increasing the threat of super-resistant bacteria emergence [2]. Antimicrobial resistance (AMR) is anticipated to cause around 10 million deaths per year by 2050, and the economic impact of AMR is expected to approach USD 100 billion during the same period [3]. It is imperative that required efforts to implement new regulations and revive research efforts to manage the AMR epidemic are carried out to address this crisis [4].

Due to the recent AMR emergence, the world is in desperate need of some relief, and to this end, the Food and Drug Administration of the United States has proposed regulations that would specify the types, quantities, and frequencies of adequate antibiotic use [5]. A complete prohibition on the use of antibiotics in cattle feed was recommended by the European Union in 2006 [6]. Japanese and Chinese policymakers, in contrast to their counterparts in Europe and the United States, have concentrated on proposals that are more compelling in nature. In 2016, the Chinese government announced the National Action Plan to Contain Antimicrobial Resistance (NAPACAR) [7]. However, despite increased awareness of antimicrobial resistance (AMR), the general situation is deteriorating, and we must continue to create antimicrobial peptides (AMPs), antibiotic combinations, and monitoring systems to effectively control AMR [8].

Artificial intelligence (AI) has demonstrated substantial competence in the field of AMR control in recent years. For example, artificial intelligence applications based on sequencing have been used to explore AMR [9]. Furthermore, the collection of clinical data for the development of clinical decision support systems could assist clinicians in monitoring trends in antimicrobial resistance to promote antibiotics’ sensible applications [10]. Additionally, artificial intelligence applications are commonly used in the development of new antibiotics and the exploration of synergistic medication combinations [11]. Interestingly, most past publications on AMR have also been written from the standpoint of structural and molecular mechanisms [12]. The schematic diagram of the possible use of AI and the dataflow integration is shown in Figure 1.

### Antibiotic Resistance; the Current Scenario

Antibiotics are medications that are utilized to both prevent and treat infections caused by bacteria and fungi in animals (significantly humans). Antibiotic resistance arises when bacteria alter their genetic makeup in response to the usage of antibiotics [13]. Antibiotic-resistant bacteria are the main causative agents of antibiotic resistance. It is possible for these germs to infect higher-order animals, and the diseases resulting from such infections are more difficult to tackle, compared with those resulting from nonresistant bacterial infections [14]. longer hospital stays, higher medical expenses, and an increased mortality rate are all associated with this phenomenon [15].

Studies show that majority of infections exhibit strong resistance to routinely used medicines; in addition, researchers are discovering gaps and breaches in surveillance and methodical data collection [16]. Based on these data, it is urged that surveillance practices must be formalized, and specific efforts must be taken to prevent AMR in the region. Furthermore, the world needs to change the way it consumes antibiotics. Without a change in public behavior, medical expeditions to find newer antibiotics will not be fruitful. The adoption of new behaviors should include measures to minimize the transmission of infectious diseases, such as immunization, hand washing, and excellent food hygiene [17].

Increasing numbers of antibiotic-resistant microorganisms are being discovered in hospitals and the general surroundings. Therefore, it is imperative to formulate new antibiotics to combat these increasing cases, but development has been slow in this area. Historically, most antibiotics have been derived from a few numbers of molecular scaffolds, with their viability extended by cycles of synthetic tailoring and optimization [18]. Considering the escalation of multidrug resistance in the most-recent generation of pathogens, the identification of novel scaffolds is a top priority. New techniques of scaffold discovery and identification are gaining traction, such as mining untapped microbial pockets for natural compounds, building screens to avoid rehashing old scaffolds, and reclaiming synthetic molecular catalogs as antibiotics [19].

Due to the development of high-throughput gene sequencing, researchers have a potent tool for profiling the complete DNA complement, which includes ARGs and DNA extracts taken from a variety of environmental sources [20]. Using this type of metagenomics technique, for example, ARGs have now been identified in many environmental samples such as soil, cattle dung, wastewater treatment plants, compost, water, and other potentially contaminated habitats [21].

## 2. Artificial Intelligence against Antibiotic Resistance

Antibiotic resistance (AMR) is, unfortunately, a result of antibiotic misuse. As AMR drastically reduces antibiotic therapeutic efficacy, it is critical that we follow its emergence and dissemination [22]. Currently, two approaches for diagnosing AMR are commonly utilized. One is called the whole-genome sequencing for antimicrobial susceptibility testing (WGS-AST) and the other one is antibiotic susceptibility testing (AST). The latter is the traditional approach for quantifying antimicrobial resistance levels, but it is not efficient, nor does it explain the mechanism of antimicrobial resistance [23]. It is possible to diagnose AMR with high accuracy and consistency using WGS-AST; however, to extract information properly, large, and high-dimensional datasets are required [12]. As a result, artificial intelligence technologies are being used to improve upon existing methodologies in the previously discussed ways [9]. Table 1 shows the application of AI in efforts to control high AMR rates with their advantages and disadvantages.

In the computer sciences field, AI has a dynamic part to play in human intelligence-stimulation systems and its research. The processes including speech recognition, visual perception, natural language processing, and decision making according to perceived data are stimulated using the technology [9,11]. The metadata from available health records and developments in processing performance are critical factors in the growth of AI systems. These two aspects are inextricably linked to complicated mathematical algorithms including neural networks (NN) and machine/deep learning, which are inextricably linked to elements such as health records and breakthroughs in computer performance [24]. This is especially true with the development of deep neural network designs, where the sophistication (commonly known as the number of factors the networks must learn) has skyrocketed in the previous decade [25]. 

ML is a part of artificial intelligence that goes through a change in its results when dealing with a large capacity of data. While specialized systems are related to the expertise of humans, likely to the human brain working. This characteristic makes it independent of man’s specialties [26]. NNs, then, are numerical informatics estimation models dependent on the working of organic neural organizations (human or creature) and, ultimately, models comprising interconnections of data that can perceive a dataset’s fundamental connections. A DNN is designed using a few layers (generally above five) of handling units that permit researchers to further develop forecasts from the information, thus finding how to comprehend them autonomously [27].

A significant advantage of NNs is that their display improves dynamically as the quantity of the dataset grows [28], allowing them to adapt to shifting information sources [11]. Currently, there are a plethora of elements associated with a patient’s consideration and clinical history that complicate patient administration. According to a new distribution, multiple times more clinical data than a person would have the choice to read in their lifetime would be provided within the present year [9]. Artificial intelligence, via naturally dealing with this massive amount of information, can assume a progressive part in supporting clinical dynamics. Nevertheless, even today, most specialists do not comprehend the convenience of AI and continue to settle on choices dependent on close-to-home insight and therapy rules [12]. This audit is to demonstrate the possible contribution of AI in battling the developing marvel of AMR, with specific pediatric patients at the center. The focus was placed on the utilization of artificial intelligence for pediatric infections in settled countries [9]. A difference between the latest AI-based diagnosis methods for AST versus the gold standards methods is shown in Figure 2.

### Assistance Strategies of AI in AMR

Early detection of infectious diseases, the differentiation between infectious and noninfectious pathologies, and correct therapy of consequences are all important aspects of combating antibiotic resistance. In this global issue, AI can play a very vital role. The preparation of antibiograms and then the development of personalized machine learning (ML)-based AMR prediction models could be very useful AI techniques for high-peak risk infectious bugs and their trends in the susceptibility patterns [9]. Using this strategy, Yelin et al. conducted a study to examine a 10-year longitudinal dataset of over 0.7 million community-acquired UTIs and identified a significant association between AMR and demographic characteristics, previous history of urine cultures, and the previous history of using the antibiotics by the patients. After examinations, they developed an ML-based AMR prediction model and described the high potential bugs for UTIs and their AMR patterns [29]. The description of the use of deep sequencing AI models is shown in Figure 3.

## 3. Use of Artificial Intelligence in Pakistan

Many studies have been performed previously to determine the prevalence of AMR in Pakistan [30,31]. However, systematic reports that provide a thorough overview of AMR in the country have not been published yet [32]. In previous studies, AMR in clinically significant bacteria in Pakistan was discussed; the authors identified the gaps in surveillance and provided references for impending work while also recommending regulations and guidelines for government officials and consumers on evidence-based measures to minimize AMR in Pakistan [13,24]. The exploitation of AI in Pakistan is not common, except in a few healthcare setups. It is somewhat unfortunate for Pakistan that only a few healthcare organizations—namely, Shaukat Khanum Memorial Cancer Hospital, Agha Khan Hospital, Shifa International Hospital, and Pakistan Kidney and Liver Institute and Research Center (PKLI&RC)—are currently implementing antibiotic stewardship program practices [33]. The Regulation and Coordination Department of the Pakistan Ministry of National Health Services, along with the NIH, pledges to develop a nationwide AMR monitoring system that includes animal, human, and resistance surveillance; creation of federal and provincial AMR labs in at least two provinces; and reduction in infection incidence and prevalence in Pakistan [31]. They also pledge to finish the project of Tricycle ESBL *E. coli* and expand the International AMR Monitoring System, as well as share relevant findings globally [13].

Pakistan is still in the early stages of implementing and applying AI systems, with very scarce available national data. One of the recent studies by Ahmed et al. (2022) focused on the knowledge, attitude, and practices of AI in the area of medicine among Pakistani medical students and professionals. Of the total studied subjects, 71.3% had a basic understanding of AI, but just 35.3% understood its subtypes, machine learning, and deep learning. The majority of research participants (77%) had no idea how AI might be used in the medical field. This demonstrates that, despite having a rudimentary understanding of AI, Pakistani physicians and medical students are unaware of its practical consequences [34]. To the best of our knowledge, the use of AI to combat such high AMR rates in Pakistan is very limited. Recently, AI facilities were used for electroencephalogram waveforms to predict failure in early school grades in children [35], as well as for childhood immunization coverage [36], mobile health applications [37], identification of risk factors for brucellosis [38], and medical diagnostics [39].

## 4. Artificial Intelligence Treating Patients in the Intensive Care Unit (ICU)

Artificial intelligence has a crucial role in ICUs, where various possibilities for implementing AI in the emergency sector have been explored. Nonadministrative AI methods have been utilized to investigate a huge amount of information stored in electronic patient data. Several AI models have been created to obtain significant data in an individual’s outline [40] and distinguish significant patient outcomes [41]. Algorithms pertaining to administered AI, given their expertise for mechanized example acknowledgment of reports, have demonstrated their applications in radiology, pathology, and histopathology [42]. Artificial intelligence is utilized widely in many medical areas in accordance with mechanical technology [43], especially in surgery and cardiology, [11] recognition of cardiac failure or arrest [44], and oncology to categorize cancer types and development stages [45].

However, with the use of AI in ICUs being in its early stages, research has effectively evaluated its utilization in the management of critical patients [46]. Many AI systems have been utilized to investigate the hospital admission duration, readmission in ICUs, mortality frequency, and the risk factors for creating unexpected infections such as sepsis. Using data from 14,480 patients, a previous study [17] developed an AI-based method to forecast patient survival and hospital admission days. The model’s area under the curve was 0.82, predicting an extended stay. This contrasts with the results of a clinical trial revealing that doctors’ accuracy in predicting ICU duration of hospital stay was around 55% [47]. A hidden Markov framework used for physiological estimates gathered during the first 48 h of ICU admission accurately predicted ICU duration of stay [48].

### Previously Adopted AI Models in ICUs in Relation to Infections and AMR

In ICUs, raw, high-dimensional inputs in form of images, numbers, text, and other data must be evaluated quickly and precisely because of the critical situation of patients. In addition, complicated, nonlinear relationships between the data must be determined. Many statistical tools have been used to represent patterns in data as mathematical equations [49]. Linear regression recommends a “best-fit line”. Deep learning (DL) does not simplify the relationship to a mathematical equation but approaches complicated medical data like a doctor would, carefully analyzing evidence to draw a reasonable conclusion. Unlike a single clinician, DL can concurrently record and evaluate several inputs, allowing prediction models to be created based on the desired result. The recurrent neural network (RNN), convolution neural network (CNN), and deep belief network (DBN) are three DL techniques that are used in ICU applications in addition to other healthcare-related AI techniques.

To predict the result of blood culture tests, Steenkiste et al. (2019) employed a temporal computational model with a bidirectional long short-term memory (LSTM) and nine clinical characteristics assessed across time from a high-quality database of 2177 ICU patients. This form of DL algorithm works effectively in situations when the time gap between an expecting event and the diagnosis is unclear. The network had an area under the receiver operating characteristic curve of 0.99 and an area under the curve (AUC) of 0.82 on average. Furthermore, the results revealed that forecasting many hours before the event is only achievable with a modest reduction in predictive power [50].

Using a dataset, Kaji et al. (2019) designed an RNN with LSTM to predict daily sepsis, myocardial infarction (MI), and administration of Vancomycin (VA) antibiotic over two weeks as the progression of patients. For sepsis, MI, and VA treatment, these models achieved the anticipated AUC of 0.823, 0.876, and 0.833, respectively. These models’ attention maps revealed the moments when input factors affected the greatest predictions, providing physicians with some interpretability. They also manifested variables that were surrogates for clinician decision making, demonstrating the difficulty of building clinical decision support systems using flexible DL techniques trained on electronic health record (EHR) data [51].

Smith et al. (2020) conducted a study to examine the possible applications of AI in microbiology and concluded that images (macroscopic or microscopic), MALDI-TOF mass spectra, and whole-genome sequences (WGS) of bacteria may all benefit from AI. In clinical microbiology, AI is starting to be applied, and it is already supporting laboratory employees with some tools for diagnostic testing. In the near future, it seems that the quantity of AI tools, the quality and reliability of AI software analyses, and the integration of AI into the clinical microbiology laboratory workflow will all increase. Microbiology technicians will increasingly depend on AI for initial screening and interpretation of routine infectious disease testing results in the future, as this allows them more time to concentrate on diagnostic problems, challenging technical interpretations, and laboratory quality control. These modifications will increase the efficiency and quality of clinical microbiology laboratory testing, benefiting both the laboratory and the patients we serve [52].

## 5. Strategies to Overcome Antibiotic Resistance

As part of their research, Getsal et al. used a combination of tools to undertake antibiotic susceptibility—namely, screening flow cytometer antimicrobial susceptibility testing and assisted machine learning were used to improve current AST methods [53]. This type of artificial intelligence technology produces a dependable output in less than 3 h. A fully developed IR-spectrometer approach has also emerged in recent years that integrates infrared (IR) spectroscopy with artificial neural networks to minimize the amount of time required to perform AST from 24 h to 30 min [54]. Pakistan, Burkina Faso, Malawi, Nepal, Bangladesh, and Zimbabwe are among the nations where the typhoid vaccine is being used, with the prospect of other countries being included during the project [55].

The antibiotic-resistant gene-sequencing models can predict antibiotic resistance categories with excellent precision (>0.97) and recall (>0.90), according to the results of an assessment of the deep and machine learning models across 30 categories of antibiotic resistance. Compared with the traditional best-hit strategy, the models demonstrated a significant benefit by consistently producing negligible false-negative results and, thus, greater total recall (>0.9) [22]. Given how neural networks underpin DeepAR Gmodels, it is reasonable to predict that the outputs of the DeepARG systems will improve even further if additional data are gathered for under-tapped categories of ARGs. DeepARG-DB, a newly constructed ARG database, contains ARGs that have been predicted with a strong confidence level and subjected to intensive manual examination, significantly increasing the scope of current ARG repositories [22]. In general, a combination of antibiotic abuse along with ineffective infection control and prevention contributes to antibiotic resistance development. Actions can be taken at all societal levels to mitigate the effects and prevent the growth of antigovernment sentiment [56].

### Strategic Considerations for Artificial Intelligence

The advancement of antimicrobial agents is, as of now, not considered a monetarily reasonable venture for the drug industries, as antibiotics are utilized for moderately brief timeframes, unlike drugs used to treat persistent infections; the improvement expenses of the latter are, along with other factors, cheaper, compared with antibiotic ventures [57]. The outcome is that, during the most recent 15 years, there have been considerable insufficiencies in the turn of events and accessibility of new antimicrobial agents to battle arising resistance cases [10]. Execution of control procedures to resolve this is a quickly developing issue, and therefore, antimicrobial stewardship is fundamental. These methodologies, although emphatically successful in adults, have been recently provided to pediatric patients [58], for which considerations of patients’ age and weight heterogeneity and designated interventions are required [9].

Consequently, in the following section, potential AI utilization against antimicrobial resistance is summarized and examined [48]. The strategies put forward in the previous section cover the following important sectors: pediatric infection prediction, analysis, and diagnosis. An important part of combating AMR is the prompt detection of infection pathologies, the separation between contagious and noninfectious pathologies, and the effective management of consequences. Children have greater infection frequencies than adults and may demonstrate nonspecific symptoms, adding to diagnostic confusion [59]. To this end, AI is a potential weapon to combat high AMR rates.

Khaledi et al. presented a technique based on support training in 2020, in which a simulated specialist learns a set of regulations from an experimental framework to enhance them and magnify the predicted outcomes [54]. This technology eliminated a patient data requirement that outperformed the practical experience of a real physician by several orders of magnitude and obtained the optimal therapy of sepsis by assessing various physicians’ viewpoints. Its application resulted in decreased mortality in patients whose physicians’ real decisions were synchronized with those of this intelligent machine, demonstrating the clinically sound potential of this technology to change sepsis therapy and assist clinicians in reaching consistent conclusions [54]. In AI’s early days, a German children’s tertiary emergency hospital undertook a trial to distinguish and assess pathogenic sepsis from nonpathogenic SIRS based on the concept that these two categories are associated with closely comparable symptoms [60]. In that trial, an ML-based diagnostic model based on an unexpected forest method was developed, considering 44 criteria available at the time of admission to the hospital (baseline characteristics, clinical/lab data, and specialized/clinical assistance). The model accounted for early recognition of all sepsis infections, and a 30 percent reduction in antibiotic use in cases of nonpathogenic SIRS was was predicted [61]. Furthermore, Liang et al. conducted a pediatric study in 2019, examining 101.6 million data points from 567,498 outpatients [40]. The main results were based on 55 diagnostic criteria that were related to prevalent pediatric disorders. Bronchopneumonia, sudden upper respiratory tract illness, bronchitis, and acute tonsillitis were among the most often detected diagnoses [40].

Nonetheless, the system performed well in detecting potentially life-threatening conditions such as meningitis. The study used logistic regression predictors to build a hierarchical health monitoring system that performed remarkably across all tissue and organ systems, demonstrating an undeniable level of superiority in predicted diagnosis, compared with original diagnoses by clinical experts [62]. These studies suggest that ML-based apps can analyze EHR in a way similar to doctors’ logical deduction and, therefore, might be used for purposes such as evaluating triage approaches or assisting doctors in the identification of difficult or unusual illnesses. The reduction in incorrect testing and expense is a considerable advantage [40].

In general, appropriate antimicrobial medication is a difficult undertaking because it involves selecting the adequate treatment for the presumed microorganism, supervising the concentration and the administration rate of the antimicrobial agent, and recognizing the optimal route to ensure the active drug levels’ turnover at the infected area [48]. It should also be highlighted in pediatrics because the types of diseases and resistance change dramatically with age, with a considerable variation in age and weight-related dosage [42]. Another problem in suggesting antimicrobials is the requirement to adjust a patient’s therapy when novel diagnostic data become available.

## 6. Artificial Intelligence Frameworks

Owing to the lack of quality healthcare resources and a large amount of information to be processed, manual observation is impractical; hence, clinicians are increasingly relying on automated decision-based supporting systems for the monitoring of antimicrobial administration [59]. To identify incorrect prescriptions and avert negative outcomes, most remedy surveillance frameworks use criteria derived from documented and expert recommendations. These frameworks are frequently insufficiently described, resulting in a high rate of clinically useless warnings. To address this issue, frameworks based on machine learning have been developed [48]. Figure 4 shows the schematic diagram of COMPOSER to predict the risk factor for sepsis.

This component, in conjunction with user feedback monitoring, was created to allow APSS to enhance its skillset over time. Anahtar demonstrated a substantial learning ability, allowing for a correct and clinically meaningful transition from parenteral to oral antibiotic therapy [42]. Mostly, in the NN literature, it is noted that the ostensibly “static” NN systems (in which the NN is prepared just once, at startup) are frequently unsuitable for purposes in which the knowledge base varies over time. Furthermore, more appropriate learning solutions have been suggested, such as reinforcement learning (as opposed to the previously stated supervised methods) or incremental digital training [42].

A comparable APSS methodology was used in another investigation to identify problematic prescribing practices that were not supported by local antibiotic stewardship professionals [10]. The learning module provided the option of removing clinically relevant requirements by identifying improper prescription that was not identified by the baseline structure [60]. Commencing with the assumption that the guideline of drug levels in the clinical outcomes of children with tuberculosis is not evident (likely because of differences in pathology configurations among youngsters and adults) and that the desired concentration for component progression is unspecified, Swaminathan et al. used a collection of AI algorithms, which include irregular forests (an ML approach executed by amassing the outcomes of isolated choice trees), to differentiate 30 clinical indicators that include research, pharmacokinetic, and therapeutic factors [40]. Along these lines, the analysts found that pharmacokinetic differences are presumably a significant contributor to treatment failure and mortality in TB patients, especially children [62].

They also identified drug concentration margins that predicted bad outcomes and discovered a negative connection between isoniazid and its companion sets—pyrazinamide and rifampicin—under specified concentration limits [55]. The challenge of healthcare costs and personnel shortages in administering appropriate antibiotic prescriptions is especially pressing in developing countries. Thus, in 2018, a group of researchers anticipated that, by employing artificial intelligence technologies to quickly acquire patient records, it is possible to achieve low-cost personalized expectations for certain antibiotics [64]. Table 2 shows the advantages, disadvantages, learning speeds, and interpretability of different AI algorithms.

## 7. Artificial Intelligence vs. Antibiotic Stewardship Program

Many publications prove the importance of an antimicrobial stewardship program (ASP). The data clearly demonstrate the significant impact an ASP has in reducing the overuse of antibiotics and reducing the collateral damage that often results from the overutilization of antimicrobials [65,66,67]. The data also indicate that an ASP leads to an infection prevention strategy [68]. The increasing risk of antimicrobial resistance due to inappropriate prescribing habits has led to devastating consequences [31]. The CDC estimated that, in the US alone, 28 lakh individuals contract an antibiotic-resistant infection, and more than 35 thousand individual annual mortality [69].

In addition, the Centers for Disease Control and Prevention also documented 223,900 incidents of Clostridioides difficile in the year 2017 and around 13,000 people were reported to be deceased [69]. As a result, the Centers for Medicare and Medicaid Services (CMS) now needed hospitals and nursing homes to implement ASPs [70]. However, this requirement is not found in the outpatient setting. Recently, the Joint Commission, an independent body that provides accreditation and certification to healthcare institutions, has made it a requirement to have ASPs in ambulatory medical institutions that are acknowledged by the joint Commission and prescribe antibiotics routinely [61]. However, this initiative has not been adopted by other agencies. Based on our review, most studies confirming the benefits of an ASP have focused exclusively on hospital settings, thereby limiting the quantity of available data regarding the benefits of an ASP in outpatient settings. Outpatient ASPs may be difficult to implement due to many obstacles [69].

This includes limited resources, such as limited personnel with expertise in infectious disease and antimicrobials, the inability to track data across multiple electronic health systems and pharmacies, as well as limited finances, support, and infrastructure [64]. Additional obstacles may include time constraints for otherwise no reimbursable tasks. Clinicians are unlikely to dedicate time to implementing an ASP if it does not generate revenue or could incur additional costs for their practice [53]. A typical ASP program may involve pharmacists, ID physicians, educational programs for providers and patients, and mechanisms in place for interventions, tracking, and reporting data. Current CMS- and relative value unit (RVU)-based payment models are tied to the number of patient visits or procedures, and therefore, dedicating time to nonpatient-specific endeavors and nonprocedural may inadvertently affect clinicians’ bottom line [22]. The CDC postulates that a minimum of 30% of prescribed antibiotics in outpatient cases are unnecessary [42]. It has published core elements to promote outpatient ASPs. However, most outpatient facilities do not have the means in place to implement these measures [56].

In addition, since medical practices have not been incentivized to participate in outpatient ASPs, this undertaking has fallen by the wayside. According to the National the Infection Prevention Strategy Ambulatory Medical Care Survey in 2016, more than 60% of patient visits in the United States were to practices with five or fewer practitioners, and only 3% of patient visits were to institutions associated with medical or academic health centers [56]. Lastly, 89.7% of patient visits were to facilities categorized as private practices. Despite the CDC’s figures on the overuse of antibiotics in outpatient settings, there have been no uniformly adapted programs in place to address the need for an ASP in outpatient settings [64]. It is, therefore, imperative that the principles of an ASP are applied to both the inpatient and outpatient settings, to truly have a significant impact on reducing the threat of antimicrobial resistance [42]. Furthermore, an ASP in the outpatient setting must be designed in a way that can be adapted by a variety of institutions with ease and efficiency, regardless of the facilities’ finances, endorsements, and resources available [3].

## 8. Conclusions

This review provides an insight into the current AI practices regarding antibiotic resistance across the world, with Pakistan the central focus. It was established that the developed nations have successfully incorporated artificial intelligence into their healthcare systems, with the most common example being electronic data entry. However, with respect to these countries, Pakistan has not yet progressed much. Only three of a multitude of Pakistani medical institutions make use of artificial intelligence, two of which are at the working stage, and one is still establishing AI systems. Moreover, none of the discussed AI frameworks is being utilized in the country, which calls for the doctors, administrators, government, and researchers’ attention as many opportunities can be tapped into for the use of AI in our systems. Global or multinational ventures can support the AI infrastructure establishment, and with continual effort, we can improve the healthcare system manyfold.

## Figures and Tables

**Figure 1 antibiotics-11-00784-f001:**
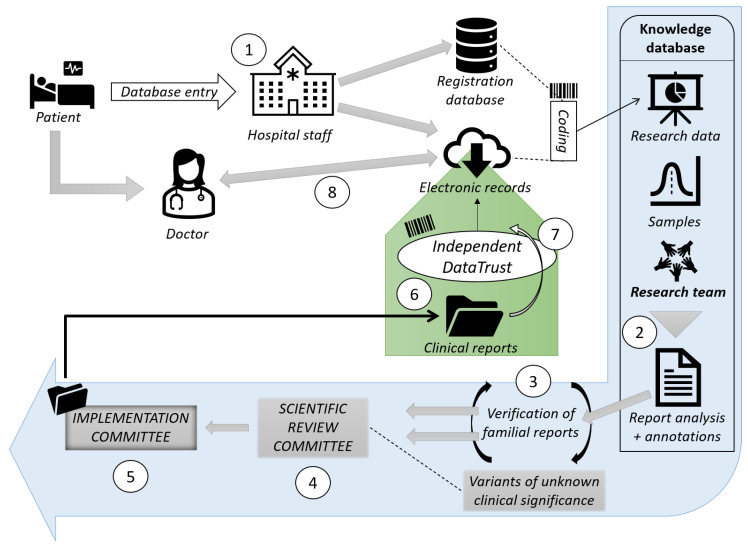
Schematic diagram of dataflow integration.

**Figure 2 antibiotics-11-00784-f002:**
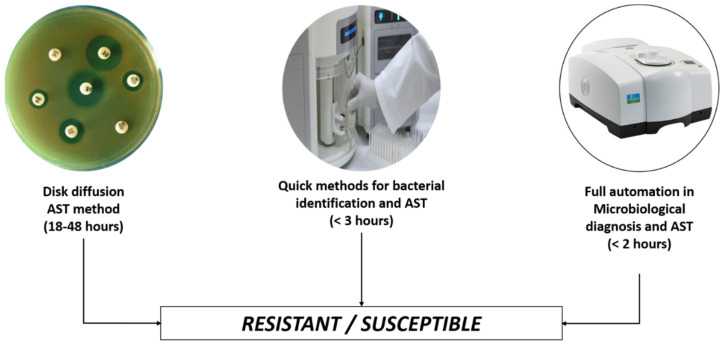
Gold standards method for AST (disc diffusion) v/s automation methods.

**Figure 3 antibiotics-11-00784-f003:**
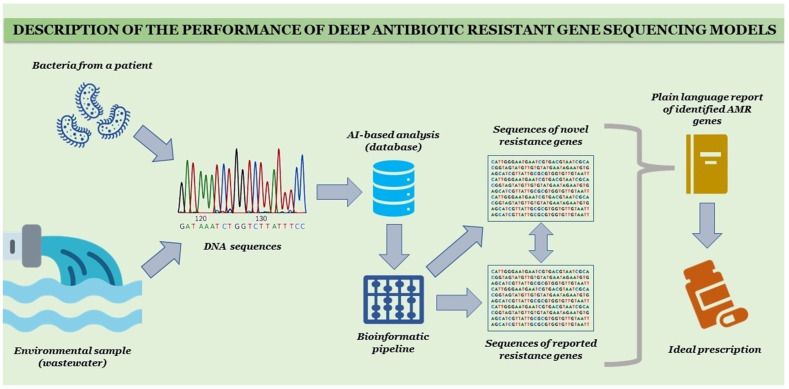
Deep antibiotic-resistant gene-sequencing model.

**Figure 4 antibiotics-11-00784-f004:**
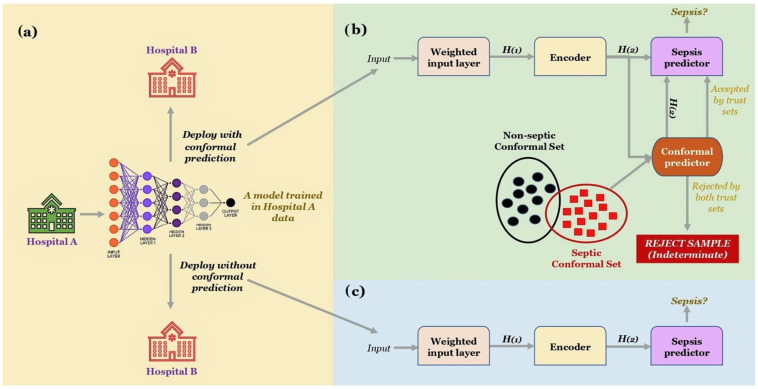
ML-based sepsis model: (**a**) during the evaluation phase of COMPOSER; (**b**) input data to obtain a risk factor for sepsis prediction; (**c**) a deployment scheme without the use of conformal prediction (adopted from Shashikumar et al. (2021) [63]).

**Table 1 antibiotics-11-00784-t001:** AI application strategies against AMR.

AI Applications for AMR	Concepts	Advantages	Drawbacks
AI health industry and antibiotics			
Antimicrobial peptides	A natural class of small host defense peptides, found in all classes of biological species.	Low chances of AMR development Multiple action mechanisms Ease of synthesis with machine/deep learning	Highly toxic Expensive in large-scale production Unpreferable widespread use The onset of allergic reactions
New antibiotics	Discovery of new and structurally different antibiotics from the ones already known using AI.	Broad-spectrum and targeted bioactivity Reduced production time Cost-effective	Challenge of training libraries according to required pharmacokinetic properties of drugs Challenge of most appropriate approach selection, minimizing toxicity, and lead compound discovery
AI, infectious diseases, and pediatric practices			
Appropriate antibiotic prescription	Appropriate therapy selection, dose, and correct administration route	Automatic support for decisions and review of antimicrobial prescriptions Automatic feedback input and relevant improvement Directed operation	Biasness in operation Little labor Need for health funds
Prediction of antibiotic resistance	ML techniques to predict early AMR or the probability of a microbial agent becoming resistant	Genomic exploitation to predict the phenotype Ability to support clinician’s decision	Lack of genotypes and genome data in NCBI or other databases Challenge of large data integration
The severity of infection prediction	Machine/deep learning tools for infectious pathology recognition and appropriate management	The efficiency of distinguishing infectious and noninfectious diseases Decision support provision Mortality reduction	Challenge of accurate data collection Insufficient relevant laboratory information

**Table 2 antibiotics-11-00784-t002:** Advantages and disadvantages of commonly used AI algorithms.

Algorithm	Description	Advantages	Disadvantages	Learning Speed	Interpretability
NB *(Naïve Bayes)*	Based on the Bayes theorem, a family of algorithms working on the principle of independent classification of each pair of features	Easily implemented, fast, suitable for missing value datasets	Independent features only	5	2
RF *(Random Forest)*	Solely based on decision trees’ predictions; takes the mean value of various trees’ outputs; precision increases with increasing no. of trees	Effective for large datasets, multi-feature handling	Insensitive to outlier information	2	3
ANN *(Artificial Neuron Network)*	Imitates the working of nerve cells in humans; makes independent judgments on new input based on learning	Multiple layer perceptron, higher accuracy with model depth	Speed of learning lowers with increasing model depth	1	1
SVM *(Support Vector Machine)*	Supervised algorithm for regression & classification; locates a hyperplane to classify data points in the N-dimensional space	Utility of kernel functions	Slow, requires specification of multiple parameters	1	1
DT *(Decision Tree)*	Prediction based on targeted variable; leaf nodes equal class label, internal node equals attributes	Easily interpreted, work with missing values in the dataset	May not work on missing data if the tree is too complex	4	5

The learning speed and interpretability increase from 1 to 5, with 5 being the best.

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
