# Peer review of "Application of Artificial Intelligence in Combating High Antimicrobial Resistance Rates"

_antibiotics, 2022, doi:10.3390/antibiotics11060784_

Round 1

Reviewer 1 Report

The manuscript entitled “Application of Artificial Intelligence in combating high Antimicrobial Resistance Rates” summaries AI practicing in antibiotic resistance area. To achieve this, the authors describe the current scenario of antibiotic resistance, the current status of AI, and how Pakistan and other counties utilize AI to help improve antibiotic resistance area, etc.

In summary, I read the paper with great interest and high expectations. I believe it is a very interesting research and story. However, this review describes different sections only in the high level, lacking the explanation of methodology and data/figure support makes this manuscript too dry to read. I recommend this paper be accepted after some major revisions.

Major revisions:

  • In section 1.2, the authors try to introduce the AI against antibiotic resistance. However, the authors use 80% of words to describe why ML/AI is importance, instead how AI is applied in antibiotic resistance area. I recommend the author exploring more how AI is applied and how AI assist in this area.
  • In section 1.3, the authors fails to explain how AI is used in Pakistan. In addition, the author do not compare with other country, even though it is mentioned in the subtitle.

  • In section 1.4, the authors compare AMR rates in different counties, I don’t think this section is related to AI or related to this paper.

  • In section 1.5, all AI applications are introduced in high level, it is hard for people to understand how AI help in the intensive care unit. I strongly recommend the authors explore more in this subsection. For example, what AI models are used, what’s the input and output for AI models, and what’s the performance (accuracy, precision, recall, f1 score, etc.) for each model. Same suggestion for other sections.
  • I strongly suggest the author adding tables and figures to support the descriptions of the performance of AI models.

Minor:

  • I don’t think there is a methodology and software contributions in this paper, however, I do see S.A, A.A.M, M.G., Y.A and U.A contribute into these two parts. I am skeptical about these contributions.
  • I recommend the author restructure this manuscript. Instead of having one section and 9 subsection under 1 section, the paper will become more organized and interesting by telling the story using multiple sections.

Author Response

Reviewer 1

Comments and Suggestions for Authors

The manuscript entitled “Application of Artificial Intelligence in combating high Antimicrobial Resistance Rates” summaries AI practicing in antibiotic resistance area. To achieve this, the authors describe the current scenario of antibiotic resistance, the current status of AI, and how Pakistan and other counties utilize AI to help improve antibiotic resistance area, etc. In summary, I read the paper with great interest and high expectations. I believe it is a very interesting research and story. However, this review describes different sections only in the high level, lacking the explanation of methodology and data/figure support makes this manuscript too dry to read. I recommend this paper be accepted after some major revisions.

Major revisions:

  • In section 1.2, the authors try to introduce the AI against antibiotic resistance. However, the authors use 80% of words to describe why ML/AI is importance, instead how AI is applied in antibiotic resistance area. I recommend the author exploring more how AI is applied and how AI assist in this area.

Response: (Line 201-216) A subsection 2.1 has been written in the revised version of manuscript.

  • In section 1.3, the authors fails to explain how AI is used in Pakistan. In addition, the author do not compare with other country, even though it is mentioned in the subtitle.

Response: (Line 217-219, 226-229, 234-248) The section 1.3 has been revised as “3. Use of artificial intelligence in Pakistan”. Furthermore, a new paragraph has been added in this section.

  • In section 1.4, the authors compare AMR rates in different counties, I don’t think this section is related to AI or related to this paper.

Response: The section 1.4 has been removed from the revised version of manuscript.

  • In section 1.5, all AI applications are introduced in high level, it is hard for people to understand how AI help in the intensive care unit. I strongly recommend the authors explore more in this subsection. For example, what AI models are used, what’s the input and output for AI models, and what’s the performance (accuracy, precision, recall, f1 score, etc.) for each model. Same suggestion for other sections.

Response: (Line 271-312) Section 4.1 has been elaborated and supported with few previously published studies.

  • I strongly suggest the author adding tables and figures to support the descriptions of the performance of AI models.

Response: (Figure 3 and 4, Table 2) 2 new figures and 1 table has been added in the revised version of manuscript to support the description of AI algorithms/models.

Minor:

  • I don’t think there is a methodology and software contributions in this paper, however, I do see S.A, A.A.M, M.G., Y.A and U.A contribute into these two parts. I am skeptical about these contributions.

Response: The author contribution section has been revised.

  • I recommend the author restructure this manuscript. Instead of having one section and 9 subsection under 1 section, the paper will become more organized and interesting by telling the story using multiple sections.

Response: The headings and subheading numbers have been revised accordingly.

Reviewer 2 Report

Antimicrobial resistance (AMR) is among the most important challenges facing Pakistan and the rest of the world. The increasing incidence of AMR has become a major problem and authorities need to take action to combat the overuse and misuse of antibiotics to combat rising resistance rates. The authors capture this through very deep documentation and presentation of the situation through AI. 
I am convinced by the whole argument, but a method, a case study on the application of successful AI should be presented. That would increase the novelty of the article.

Author Response

Reviewer 2

Comments and Suggestions for Authors

Antimicrobial resistance (AMR) is among the most important challenges facing Pakistan and the rest of the world. The increasing incidence of AMR has become a major problem and authorities need to take action to combat the overuse and misuse of antibiotics to combat rising resistance rates. The authors capture this through very deep documentation and presentation of the situation through AI. I am convinced by the whole argument, but a method, a case study on the application of successful AI should be presented. That would increase the novelty of the article.

Response: (Line 236-248, 271-312) Dear reviewer, we would like to pay our sincere thanks to you for your kind consideration of the current manuscript to review and we appreciate your comment and suggestion to add one case study. Unfortunately, at this time we don’t have the possible case report to be included in the current review. Other then to add the original case report, we have added some previously published original studies to support our statements. Furthermore, 2 new figures and 1 table (Figure 3 and 4, Table 2) has been added in the revised version of manuscript to elaborate the AI models.

Reviewer 3 Report

The study aims to emphasize and review the application of Artificial Intelligence in mitigating the  increasing rate of Antimicrobial resistance (AMR) in developing countries. While the topic is interesting; the article does not deliver on its title. The review of  AI models  presented in the paper is not coherent and the writing quality  needs significant improvement. The paper scrambles several references without properly organizing and categorizing them. For example, section 1.5 cites several studies that applied AI to forecast patients' survival and length of stay in ICU but it fails to  mention how this is related to the AMR problem which is the focus of the paper.   Similarly, sections 1.6.1-1.6.3 give some recommendations for managing antibiotic resistance at individual and administrative levels which again has nothing to do with AI and its application in AMR. 

Authors are recommended to organize the literature by 1- the specific type of AI application in AMR and 2- discuss each AI model that is reviewed in more depth ( for example, explain what type of AI model was used in each application and what type of data it has been trained on)

In addition, the manuscript contains several tables and figures without referencing them in the text.

Author Response

Reviewer 3

Comments and Suggestions for Authors

The study aims to emphasize and review the application of Artificial Intelligence in mitigating the increasing rate of Antimicrobial resistance (AMR) in developing countries. While the topic is interesting; the article does not deliver on its title. The review of AI models presented in the paper is not coherent and the writing quality needs significant improvement. The paper scrambles several references without properly organizing and categorizing them.

For example, section 1.5 cites several studies that applied AI to forecast patients' survival and length of stay in ICU but it fails to mention how this is related to the AMR problem which is the focus of the paper. 

Response: (Line 371-312) Section 4 has been elaborated and supported with few previously published studies. A subsection “4.1. Previously adopted AI models in ICUs in-relation to infections and AMR” has been added in the revised version of manuscript.

Similarly, sections 1.6.1-1.6.3 give some recommendations for managing antibiotic resistance at individual and administrative levels which again has nothing to do with AI and its application in AMR. 

Response: Subsection 1.6.1-1.6.3 has been removed from the revised version of manuscript.

Authors are recommended to organize the literature by:

  1. the specific type of AI application in AMR and discuss each AI model that is reviewed in more depth (for example, explain what type of AI model was used in each application and what type of data it has been trained on)

Response: (Figure 3 and 4, Table 2) 2 new figures and 1 table has been added in the revised version of manuscript. Furthermore, the manuscript has been revised and the description about AI models has been provided in the revised version of manuscript.

In addition, the manuscript contains several tables and figures without referencing them in the text.

Response: The figures and tables has been carefully checked and cited in the text accordingly.

Round 2

Reviewer 3 Report

Authors adequately addressed the issues raised in the first review round. I recommend that authors proof read their manuscript before publication.

Author Response

Reviewer 2

Comments and Suggestions for Authors

Authors adequately addressed the issues raised in the first review round. I recommend that authors proof read their manuscript before publication.

Response: Dear Reviewer, we would like to say our sincere thanks to you for your kind efforts to our manuscript. After addressing your comments, the manuscript has become better for the readers. Furthermore, the manuscript has been thoroughly revised for English proofreading.
